# Sonoporation Using Nanoparticle-Loaded Microbubbles Increases Cellular Uptake of Nanoparticles Compared to Co-Incubation of Nanoparticles and Microbubbles

**DOI:** 10.3390/pharmaceutics13050640

**Published:** 2021-04-30

**Authors:** Sofie Snipstad, Sigurd Hanstad, Astrid Bjørkøy, Ýrr Mørch, Catharina de Lange Davies

**Affiliations:** 1Department of Physics, Norwegian University of Science and Technology, Høgskoleringen 5, 7491 Trondheim, Norway; sigurd.hanstad@gmail.com (S.H.); astrid.bjorkoy@ntnu.no (A.B.); catharina.davies@ntnu.no (C.d.L.D.); 2Department of Biotechnology and Nanomedicine, SINTEF Industry, Sem Sælandsvei 2A, 7034 Trondheim, Norway; yrr.morch@sintef.no; 3Cancer Clinic, St. Olav’s Hospital, Prinsesse Kristinas Gate 1, 7030 Trondheim, Norway

**Keywords:** drug delivery, cancer, microbubbles, nanomedicine, sonopermeation, sonoporation, ultrasound

## Abstract

Therapeutic agents can benefit from encapsulation in nanoparticles, due to improved pharmacokinetics and biodistribution, protection from degradation, increased cellular uptake and sustained release. Microbubbles in combination with ultrasound have been shown to improve the delivery of nanoparticles and drugs to tumors and across the blood-brain barrier. Here, we evaluate two different microbubbles for enhancing the delivery of polymeric nanoparticles to cells in vitro: a commercially available lipid microbubble (Sonazoid) and a microbubble with a shell composed of protein and nanoparticles. Various ultrasound parameters are applied and confocal microscopy is employed to image cellular uptake. Ultrasound enhanced cellular uptake depending on the pressure and duty cycle. The responsible mechanisms are probably sonoporation and sonoprinting, followed by uptake, and to a smaller degree enhanced endocytosis. The use of commercial Sonazoid microbubbles leads to significantly lower uptake than when using nanoparticle-loaded microbubbles, suggesting that proximity between cells, nanoparticles and microbubbles is important, and that mainly nanoparticles in the shell are taken up, rather than free nanoparticles in solution.

## 1. Introduction

Ultrasound contrast agents are gas microbubbles with a shell of lipid, polymer or protein [1,2] that are routinely used for diagnostic ultrasound imaging [3,4]. Recently, microbubbles have also been used in therapeutic applications and have shown promising results in ultrasound-mediated delivery of drugs to tumors and across the blood-brain barrier [4]. Ultrasound can cause heating of tissue or mechanical effects such as acoustic radiation force and cavitation [5]. Cavitation is induced when an ultrasound field causes the microbubble to undergo volumetric oscillations and/or collapse. The resulting biophysical effects include pushing and pulling on nearby surfaces as the bubbles vibrate, microstreaming in the surrounding fluid leading to shear stress on nearby surfaces, or shock waves and jet streams as microbubbles collapse [1,3,6,7,8,9,10,11,12,13]. Acoustic radiation forces can also cause translocation of microbubbles [1,3,10,12,14,15]. Altogether, these effects can increase the permeability of tissue, leading to improved accumulation and penetration of drugs, antibodies, genes, cells and more [5,10,12,16,17] in tumors and across the blood-brain barrier [18].

In vitro, the biophysical effects of microbubbles are reported to cause sonoporation [2], creating transient pores in the cell membrane into which drugs can diffuse [13,19,20,21,22], to enhance endocytosis resulting in increased active uptake of drugs [23,24,25], and to perturb the organization of the cellular cytoskeleton [26]. The majority of in vitro experiments have been performed with a co-incubation of commercial diagnostic microbubbles [16] such as Sonovue [23], Sonazoid [27], Targestar [28,29], Definity [30] or Optison [30] and small molecules like dextrans [23,24,30], propidium iodide [13,19,20,26,28,29,30] or calcein [28], with sizes ranging from a few to tens of nanometers. In recent years, various forms of custom-made microbubbles have been developed for therapeutic purposes [1,3]. Loading the therapeutic agent onto the microbubbles has resulted in more efficient delivery both in vivo [31,32] and in vitro to cells and spheroids [33,34,35,36], especially for larger therapeutics such as drug-loaded nanoparticles. Recently, the underlying mechanism was suggested to be a phenomenon called sonoprinting, where the nanoparticles are deposited in patches on the cell membrane before being internalized [33,34,35]. Such an approach allows for controlled delivery in a spatiotemporal manner and is especially promising for drugs that benefit from encapsulation in nanocarriers due to reduced premature degradation, increased solubility, improved pharmacokinetics and biodistribution, targeted delivery and reduced toxicity, increased cellular uptake and prolonged release properties.

We have previously shown that nanoparticle-loaded microbubbles could improve the delivery and therapeutic effect of nanoparticles in various tumor models in mice [37,38,39], and also demonstrated that the same platform could be used to locally and reversibly open the blood-brain barrier for the delivery of nanoparticles to the brain [40,41,42]. When injected intravenously, the nanoparticle-loaded microbubbles are constrained to the vasculature, and will upon ultrasound exposure interact with the endothelial cells, enabling nanoparticles to extravasate either paracellularly or transcellularly. To obtain knowledge on possible uptake into cells, the present study investigates to what extent nanoparticle-loaded microbubbles can enhance nanoparticle delivery into cells in vitro by sonoporation or enhanced endocytosis. This work was encouraged by the paper of De Cock et al. [33] presenting the concept of sonoprinting. Using nanoparticle-microbubble complexes, they demonstrated that the proximity between nanoparticle, microbubble and cell surface is crucial for successful cellular uptake. We hypothesized that also our nanoparticle-loaded microbubble would enhance the cellular uptake of nanoparticles when compared to the co-incubation of nanoparticles and microbubbles. To study the impact of proximity between microbubbles and the relatively large nanoparticles, we compare the nanoparticle-loaded microbubbles to a commercially available lipid microbubble co-incubated with nanoparticles. There are few studies demonstrating sonoporation and cellular uptake of nanoparticles, mainly cellular uptake of smaller molecules has been shown. Understanding the underlying mechanisms is key for the evolution of sonoporation as a biomedical tool, especially for nanoparticles. Various ultrasound parameters are applied (varying pressure, pulse length and pulse repetition frequency) to understand how these parameters affect delivery. Confocal laser scanning microscopy (CLSM) is employed to image intracellular and membrane-associated fluorescently labeled polymeric nanoparticles. Experiments are done at 37 °C and 4 °C to distinguish between sonoporation/sonoprinting and endocytosis [43].

## 2. Materials and Methods

### 2.1. Cell Culture

Human prostatic adenocarcinoma (PC3) cells (American Type Culture Collection, Manassas, VA, USA) were cultured in Dulbecco’s Modified Eagle Medium (DMEM, Gibco Thermo Fischer Scientific, Waltham, MA, USA) supplemented with 10% fetal bovine serum (FBS, Sigma-Aldrich, St. Louis, MO, USA) and 1% penicillin-streptomycin (100 U/mL and 100 µg/mL, Sigma-Aldrich). The cells were maintained in exponential growth at 37 °C with 5% CO_2_. Before every experiment, the cells were detached by trypsination, counted and resuspended in medium. In 10 mL of medium, 400,000 to 500,000 cells were seeded in CLINIcell culture chambers (Mabio, Tourcoing, France). The CLINIcell has two optically and acoustically transparent polycarbonate gas permeable membranes with areas of 25 cm^2^ and plasma treated surfaces to promote cell adhesion [41]. The cells were allowed to grow for four days to obtain a nearly confluent layer on the day of the experiment.

### 2.2. Synthesis of Polymeric Nanoparticles

To synthesize poly(2-ethyl butyl cyanoacrylate) (PEBCA) nanoparticles covered with poly(ethylene glycol) (PEG), a one-step mini-emulsion polymerization was performed as previously described [44]. Briefly, a water phase containing Brij L23 (9 mM, 23 PEG units, MW 1225, Sigma-Aldrich) and Kolliphor HS15 (11 mM, 15 PEG units, MW 960, Sigma-Aldrich) in 0.1 M HCl was mixed with an oil phase consisting of the monomer 2-ethyl butyl cyanoacrylate (Henkel Biomedical, Düsseldorf, Germany), 1.7 wt% Miglyol 812 (co-stabilizer, Cremer, Cincinnati, OH, USA), 0.81 wt% V65 (Azobisdimetyl valeronitril, Wako, Osaka, Japan), 0.1 wt% methanesuflonic acid (Sigma-Aldrich) and 0.5 wt% of the fluorescent dye NR668 [43,45]. This hydrophobic dye has previously been shown to be stable and not leak out of the nanoparticles [43]. The mixture was sonicated for 3 min on ice (50% amplitude, Branson Ultrasonics digital sonifier 450, Danbury, CT, USA). The solution was kept on rotation (15 rpm) for one hour at room temperature before adjusting the pH to 5 using 0.1 M NaOH. Polymerization was continued overnight at room temperature on rotation. The dispersion was dialyzed (Spectra/Por dialysis membrane MWCO 12-14000 Da, Spectrum Labs, Rancho Dominguez, CA, USA) against 1 mM HCl to remove unreacted PEG and dye. The size, zeta potential and size distribution of the nanoparticles were measured by dynamic light scattering (DLS, Zetasizer, Malvern Instruments, Malvern, UK). The nanoparticles were previously also characterized with respect to size, charge, morphology, PEGylation, stability of fluorescent payload, cellular uptake, degradation and toxicity [43,44,46,47,48].

### 2.3. Synthesis of Microbubbles

Two types of microbubbles were used. The first was nanoparticle-loaded microbubbles, which were made by mixing 1% *w*/*v* PEBCA nanoparticles with the protein casein (0.5% *w*/*v*) in 0.9% phosphate-buffered saline (PBS). The solution was saturated with perfluoropropane (F2 Chemicals, Preston, Lancashire, UK) before mixing with an Ultra Turrax (24,000 rpm, Branson Ultrasonics) for 4 min. Both the protein and the nanoparticles are needed for the successful formation of microbubbles; the shell self-assembles upon vigorous stirring. The resulting microbubbles were imaged in Countess cell counting chambers (Thermo Fisher Scientific) and analyzed in ImageJ (National Institutes of Health, 1.48 v, Bethesda, MA, USA) to determine their size and concentration. In addition to the nanoparticles on microbubbles, the resulting solution also contained an excess of free nanoparticles. The microbubbles have previously been characterized with respect to concentration, size distribution, stability in vitro and in vivo, reproducibility and acoustic properties in previous publications [37,38,39,40,41,44]. Electron microscopy imaging has indicated that nanoparticles form a continuous monolayer on the microbubble surface with varying shell thickness [44]. To visualize their morphology, the microbubbles were sputter-coated with 5 nm gold (Cressington 308R, Cressington Scientific Instruments Ltd., Watford, UK) and imaged by scanning electron microscopy using an S5500 S(T)EM (Hitachi High-Tech Corporation, Tokyo, Japan). To visualize their nanoparticle loading, the microbubbles were also imaged with confocal microscopy, using the same microscope and settings as described for the cells below.

The second type of microbubbles used was Sonazoid, a lipid bubble containing perfluorobutane stabilized by hydrogenated egg phosphatidylserine (HEPS). The Sonazoid were prepared according to the supplier’s (GE Healthcare, Chicago, IL, USA) recommendations. A vial of freeze-dried microbubbles, 2.4 × 109 in total, were diluted with 2 mL of PBS, and the solution homogenized by careful flipping of the vial for 60 s. Sonazoid has a median diameter of approximately 2.6 µm [49].

### 2.4. Experimental Setup and Ultrasound Treatment

The experiments were performed in a water tank with a transducer mounted in the bottom and CLINIcell mounted at a distance of 125 mm, to make sure the cells were in the focal position of the ultrasound beam. The transducer was characterized in a water tank system (AIMS-III, Onda, Sunnyvale, CA, USA) with a calibrated hydrophone (HGL-0200, Onda), and the -3dB beam width at the target was 3 mm. The water tank was filled with degassed water and placed in a room with a constant temperature of 37 °C or 4 °C. An absorbing lid was placed on the top of the tank to avoid standing waves. A signal generator (33500B Series Waveform Generator, Agilent, Santa Clara, CA, USA), an amplifier (2100L-1911 RF Power Amplifier, ENI, Rochester, NY, USA) and a custom-made single element-focused ultrasound transducer with a center frequency of 1 MHz (Imasonic, Besancon, France) were connected in series to produce the ultrasound signal. Pulsed waves were transmitted at peak negative pressures of 0.1, 0.2, 0.32, 0.45, 0.59 and 0.85 MPa (mechanical index (MI) of 0.1–0.85). Two different exposure schemes were used: a pulse repetition frequency (PRF) of 50 Hz was applied with pulses of 100 cycles, resulting in a duty cycle (DC) of 0.5%, or a PRF of 100 Hz was applied with pulses of 1000 cycles, resulting in a DC of 10%. The total treatment time was 1 min.

### 2.5. Sonoporation Experiments

All chemicals were heated to 37 °C or cooled to 4 °C before use. The cell medium was carefully removed from the CLINIcell and replaced with 10 mL of medium containing the microbubbles and nanoparticles. The CLINIcell was then flipped upside down and placed in the tank for 30 min to allow the microbubbles to rise towards the cell monolayer, to achieve close contact between the cells and the microbubbles, as illustrated in Figure 1. Four different areas of the CLINIcell were then treated with ultrasound. The CLINIcell was then removed from the tank and the medium was replaced with medium containing 2.5 μg/mL Cell Mask Deep Red plasma membrane stain (Thermo Fischer Scientific), to stain the cell surface. After 5 min, the medium was again removed from the CLINIcell and replaced with PBS (Sigma-Aldrich), before imaging could begin.

Two different microbubbles were compared: Sonazoid that were co-incubated with polymeric nanoparticles, and microbubbles with a shell composed of protein and polymeric nanoparticles co-incubated with nanoparticles in solution. The same concentrations of microbubbles and nanoparticles were used in both cases. A concentration of about 10 microbubbles per cell was added to the CLINIcell, corresponding to about 15 million microbubbles for a confluent CLINIcell. This was found, through optical microscopy, to be a suitable number of microbubbles, ensuring that most cells were in contact with one or a few microbubbles, without leading to microbubble aggregates. Control CLINIcells were also included and underwent the exact same procedure, except that ultrasound was not turned on.

### 2.6. Confocal Imaging

Imaging the cells started directly after ultrasound treatment and Cell Mask staining were done. A confocal laser scanning microscope (LSM800, Zeiss, Oberkochen, Germany), equipped with water-immersion apochromat objectives (40×/1.2 and 25×/0.8) was used. A 488 nm laser excited the NR668 of the nanoparticles and a 640 nm laser excited the Cell Mask stain, and fluorescence was detected at 590–700 nm and 645–700 nm, respectively. Laser intensity and detector gain were adjusted to obtain maximum signal with minimum background. The treated areas of the CLINIcell were examined through the oculars, and z-stacks were acquired for cells of interest that showed uptake of particles. The location of the nanoparticles relative to the cell surface was evaluated through reconstitution of a 3D image. Nine z-stacks were captured at each treatment area in a 3 × 3 grid spaced 1 mm apart. Similarly, control images were taken at untreated areas of the CLINIcell. Approximately 50–80 cells were imaged per stack when the 25× water lens was used.

### 2.7. Image and Data Analysis

Image analysis of the z-stacks was performed in MatLab (custom-made scripts, The MathWorks, Natick, MA, USA). The intensity threshold value for the Cell Mask channel was determined by calculating the average intensity in a background region of interest. For the nanoparticle channel, the threshold was determined by Otsu’s method, applied to the maximum projection image. A binary image from a nanoparticle z-stack is shown in Figure 2A. Pixels corresponding to the nanoparticles are white. The corresponding binary image of the Cell Mask stain is shown in Figure 2B. In Figure 2C, all pixels inside each cell in addition to the cell surface have been set to white and objects smaller than cells have been removed. The binary stack (3D image) was used to determine if each nanoparticle spot was inside or outside a cell, or colocalized with the cell membrane (the borders in the binary Cell Mask image, Figure 2B). For quantification, all voxels with a nanoparticle signal and all voxels with an intracellular or cell membrane signal in the z-stack were counted. The total number of nanoparticle voxels was then normalized to the total number of cell voxels or cell surface voxels. Data from the different images of each treatment area were averaged and the standard deviation was calculated. Statistical analysis was performed using R and XLSTAT in Microsoft Excel. The data were first tested for normality with multiple Shapiro-Wilk tests. For some groups, the data were not univariate normal. A non-parametric method, pairwise Wilcoxon rank-sum tests (Mann-Whitney U tests), was performed to test for significance between the different treatments and the untreated control. A significance level of 0.05 was used in all hypothesis tests. To correct for multiple comparisons, the false discovery rate was used to adjust the *p*-values.

## 3. Results

The average nanoparticle diameter was 177 nm (z-average) with a polydispersity index of 0.17 and a zeta potential of −1.3 mV. The average diameter of the nanoparticle-loaded microbubbles was 2.5 μm (Figure 3) and the concentration 3 × 10^8^ microbubbles/mL. Representative electron microscopy and confocal microscopy images of the microbubbles demonstrate their morphology and nanoparticle loading on the microbubble surface (Figure 3), along with the size distribution of the microbubbles.

Cells exposed to nanoparticle-loaded microbubbles and ultrasound showed higher uptake of nanoparticles than cells exposed to ultrasound in the presence of Sonazoid and nanoparticles. The number of nanoparticles taken up in each cell varied, as shown in Figure 4 and Figure 5. Both individual nanoparticles and groups of a few, as well as larger clusters of nanoparticles, were internalized. For the cells treated with Sonazoid and ultrasound, individuals and groups of a few nanoparticles were mainly seen (Figure 5B). Clusters of nanoparticles were observed less frequently than for the cells treated with nanoparticle-loaded microbubbles (Figure 5C). Nanoparticles were observed both inside the cells and on the cell membrane colocalizing with the Cell Mask staining (Figure 5D). The majority of the cells did not show any nanoparticle accumulation. For the untreated control cells, usually, very few nanoparticles were observed per cell, and fewer cells had nanoparticles than in the ultrasound-treated cell population (Figure 5A).

Quantification of the cellular uptake of nanoparticles confirmed that ultrasound treatment combined with nanoparticle-loaded bubbles resulted in a more efficient delivery of nanoparticles to the cells compared to the commercial lipid microbubbles (Figure 6). Both microbubbles in combination with focused ultrasound improved cellular uptake compared to untreated cells (1.2–1.7 fold increase for Sonazoid, 2.8–4.4 fold increase for nanoparticle-loaded bubbles). The large standard deviation reflects the large variety of cellular uptake of nanoparticles, where some cells have several aggregates of nanoparticles and the majority of cells have no nanoparticles. Cells incubated with the nanoparticle-loaded bubbles at 4 °C and exposed to ultrasound showed considerable fluorescence (1.6–2.8 fold increase compared to untreated control), however, the uptake at 37 °C was higher than at 4 °C (1.1–1.7 fold higher). For the nanoparticle-loaded microbubbles at both temperatures, cellular uptake increased with increasing ultrasound pressure (MI from 0.32 to 0.59) when pulse length and PRF were kept constant at 100 cycles and 50 Hz, respectively. Maximum cellular uptake was obtained for MI = 0.59 and increasing the MI to 0.85 did not increase cellular uptake further. Sonazoid in combination with ultrasound did not increase the cellular uptake of nanoparticles with increasing MI.

To determine the lower limit of acoustic pressure needed to increase cellular uptake, MIs of 0.1 and 0.2 were also applied for the nanoparticle-loaded microbubbles. Ultrasound with an MI of 0.1 did not increase cellular uptake significantly compared to the untreated control cells. However, a clear increase in uptake was observed for all MIs above 0.1 (Figure 7A), indicating a threshold for delivery between 0.1 and 0.2. Increasing the pulse length from 100 to 1000 cycles and the PRF from 50 Hz to 100 Hz (DC increased from 0.5% to 10%) increased the cellular uptake by approximately 50%.

The number of nanoparticles that colocalized with the cell membrane was more pronounced for lower MIs (Figure 7B) and higher DCs (Figure 7D).

## 4. Discussion

### 4.1. Comparison of Microbubbles

Ultrasound in combination with microbubbles was found to increase the uptake of nanoparticles, and nanoparticle-loaded microbubbles were significantly more effective than commercial lipid Sonazoid microbubbles. This is probably due to the proximity between the microbubbles, nanoparticles and cells [33]. As nanoparticle-loaded microbubbles oscillate and collapse, their shell fragments into individual nanoparticles and shell fragments consisting of multiple nanoparticles [50], which are then subsequently delivered to the cell. If the nanoparticles are to be taken up through pores created by sonoporation, they need to be present at the pore openings immediately after they occur since such pores are transient and close rapidly, usually within a few seconds and after no more than a couple of minutes [10,13,20,51,52,53]. In the case of Sonazoid, even though the total number of available nanoparticles is the same as for the nanoparticle-loaded microbubbles, all nanoparticles are free in suspension. Some of these are delivered to the cells, however, the cavitation of lipid microbubbles and resulting microstreaming [9,54,55] have been reported to push free nanoparticles away from the cells [33,35], explaining the less efficient delivery. Our findings are consistent with similar observations using liposomes attached to microbubbles [33,34,35]. Microbubbles loaded with nanoparticles in combination with ultrasound increased the uptake of nanoparticles in cells [33] and spheroids [34], and improved the therapeutic effect in vivo [31,32].

Another explanation for the more effective delivery using nanoparticle-loaded microbubbles could be that the two types of microbubbles respond differently to the ultrasound wave due to their different shell properties. Sonazoid has a thin and flexible lipid shell, while nanoparticle-loaded microbubbles have a more rigid, thicker and irregular shell consisting of proteins and nanoparticles [39,44]. As described by Kooiman et al., flexible bubbles will have different oscillation regimes compared to rigid ones [1]. The flexible lipid-shelled microbubbles will undergo oscillations at low pressures [56]. Hard-shelled microbubbles will not respond so willingly to the applied ultrasound wave; they remain unaffected at low pressures but display destruction with shell rupture and gas release at higher pressures [57]. Comparing the cavitation signals from oscillating nanoparticle-loaded microbubbles and lipid microbubbles (SonoVue) showed that the cavitation signal from nanoparticle-loaded microbubbles lasted for a longer time [39]. Loaded microbubbles were also reported to have a higher pressure threshold for the onset of microbubble oscillations [58]. If the nanoparticle-loaded microbubbles collapse in a more violent manner than the lipid microbubbles, this could also lead to larger pores in the cells, and thus, room for more material to be delivered. Consistent with this, larger clusters of nanoparticles were localized intracellularly using nanoparticle-loaded microbubbles.

The response of the microbubbles depends on applied ultrasound parameters, but also on microbubble size [57,59] and composition [10], and it is known that a certain radial expansion is needed for successful sonoporation [20]. Due to the heterogeneous shell in terms of thickness and nanoparticle coverage [44], in combination with polydispersity [39] within a batch, we expect that different microbubbles within a batch will show variable behavior, and thus, variable cell interaction at the same ultrasound settings. This could, in combination with various delivery mechanisms, explain the large variation in uptake observed between individual cells. Making microbubbles monodisperse to have a more homogeneous response to ultrasound treatment could probably lead to more effective drug delivery [1,35]. The majority of cells did not contain nanoparticles. The lack of nanoparticle uptake in cells might be due to no interaction between the cells and nanoparticle-loaded microbubbles, demonstrating the need for proximity between nanoparticles, oscillating microbubbles and cells, as previously described also for liposomes linked to microbubbles [33,34,35], and the importance of optimizing both the type of microbubble and the type of ultrasound treatment to make the delivery as efficient as possible.

### 4.2. Effect of Ultrasound Parameters

The cellular uptake after treatment with nanoparticle-loaded microbubbles increased with increasing MI above 0.1, which is in the range described previously [29]. Above this threshold, the pores created by sonoporation seem to be large enough for nanoparticles and even clusters of nanoparticles to enter the cells. When MI increases and the bubbles collapse, smaller shell fragments can be formed, as was observed for lipid bubbles [60]. Increasing acoustic pressures generally lead to a larger microbubble response and larger pores in the cell membranes, increasing uptake with increasing MI [24,29,30,61,62], which was also shown in vivo [63]. Pore sizes as large as 30 μm^2^ were observed to reseal successfully [53], allowing potential aggregates of nanoparticles to enter the cell.

Furthermore, increasing the DC (by increasing the pulse length and PRF) increased the number of nanoparticles taken up in the cells, and especially the number of nanoparticles colocalized with the cell membrane. This could be an effect of the increased radiation force associated with increasing pulse length and PRF, leading to translational movement of microbubbles [28,64] and pushing microbubbles and nanoparticle aggregates against the cell membrane, promoting adhesion to the cells or embedding in the cell membrane. Increasing the DC also increases the amount of aggregation due to secondary Bjerknes forces [65], leading to larger aggregates of bubbles, which will be more heavily affected by primary radiation forces [14,66,67], potentially increasing the number of particles attached to cell membranes. Increasing the PRF was also found to have the same effect as increasing the MI, leading to larger membrane pores and increasing delivery [62].

### 4.3. Delivery Mechanisms

The responsible mechanisms for the increased cellular delivery after treatment with nanoparticle-loaded microbubbles and ultrasound were probably sonoporation and sonoprinting, and to a smaller degree enhanced endocytosis. The considerable uptake at 4 °C suggests that endocytosis was not the main mechanism. Sonoprinting was recently suggested as an uptake mechanism for liposomes attached to microbubbles [33,34,35]. Roovers et al. [35] found that non-spherically oscillating microbubbles released their nanoparticle payload in the first few cycles of ultrasound insonation. At low acoustic pressures, the released nanoparticles were then transported away from the cells by microstreaming, which does not favor uptake of nanoparticles by the cells. However, higher acoustic pressures (>300 kPa) and longer ultrasound pulses (>100 cycles) lead to rapid translation of the microbubbles due to acoustic primary and secondary radiation forces. As a result, the released nanoparticles were transported along in the wake of the microbubbles, which eventually led to the deposition of nanoparticles in elongated patches on the cell membrane, i.e., sonoprinting. They also concluded that the effect was more pronounced with increasing pressures and pulse lengths [35]. In the present study, deposition of nanoparticles onto the cell membrane was observed, indicating that sonoprinting is involved in the delivery from nanoparticle-loaded microbubbles. Sonoporation and sonoprinting are difficult to distinguish without time-resolved imaging during and after treatment. However, the finding that the nanoparticle shell seems so important for the uptake supports the finding from De Cock et al. that sonoprinting is an important mechanism of delivery from nanoparticle-loaded bubbles to cells in vitro [33].

### 4.4. Clinical Relevance

Since microbubbles are generally too large to escape from the blood vessel, they will not be in contact with the cancer cells in vivo. Some alternative approaches already exist, including phase shifting nanoemulsions or nanobubbles, which could extravasate and elicit a direct effect on cancer cells in tissue [68,69,70,71]. Recently, it was also demonstrated that sonoporation can indirectly induce intercellular gaps between neighboring cells [20], which remain open for tens of minutes, allowing delivery of therapeutic agents beyond the vascular endothelial cell layer, for instance, in tumors or across the blood-brain barrier.

Applications within cardiovascular pathologies would also be interesting [19,67] since the endothelial cells are in direct contact with the microbubbles, for instance, in gene delivery to induce angiogenesis to stimulate reperfusion after myocardial infarct [72,73]. As previously suggested, immune cells would be another interesting target in vivo, to which the microbubbles can have direct contact [33]. Dewitte et al. used lipid microbubbles loaded with mRNA-lipoplexes for in vitro sonoporation of dendritic cells for a cancer vaccine, and showed effective induction of antigen-specific T-cells in vivo, resulting in reduced tumor growth and long-term immunological memory [74]. Encapsulation in nanoparticles can be highly beneficial for certain components used in gene therapy and immunotherapy, and applying ultrasound for sonoporation could be an important way of enhancing delivery in such applications in the future.

## 5. Conclusions

Sonoporation by co-administration of microbubbles and drugs has been shown to be efficient for small molecules, but not for delivering larger nanoparticles to cells. Nanoparticle-loaded microbubbles in combination with ultrasound enhanced the cellular uptake of nanoparticles more efficiently than Sonazoid and ultrasound, suggesting that proximity between cells, nanoparticles and oscillating microbubbles is essential. The responsible mechanisms were probably a combination of sonoporation and sonoprinting, followed by cellular uptake, and to a smaller degree enhanced endocytosis. This is relevant for enhancing the delivery of various drugs and genes, and could potentially be used in applications within the treatment of cancer, brain diseases, immunological diseases and cardiovascular pathologies.

## Figures and Tables

**Figure 1 pharmaceutics-13-00640-f001:**
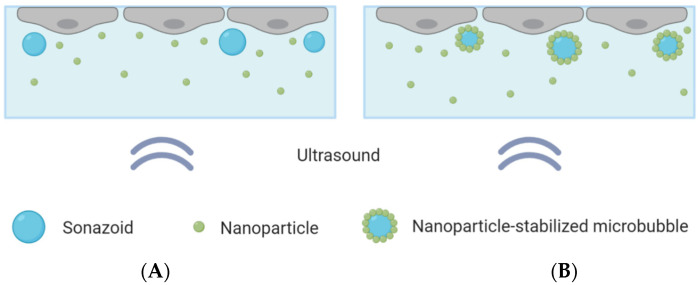
Schematic overview of the CLINIcell with a cell layer on top, microbubbles and nanoparticles in solution and ultrasound applied from below. (**A**): Cells exposed to nanoparticles co-incubated with Sonazoid. (**B**): Cells exposed to nanoparticle-loaded microbubbles and an excess of nanoparticles.

**Figure 2 pharmaceutics-13-00640-f002:**
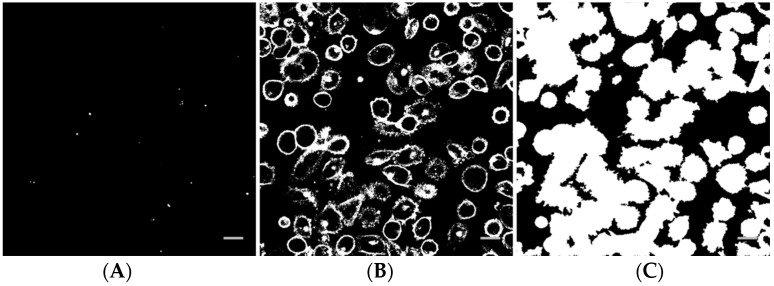
Example of a binary nanoparticle image (**A**), cell surface image (**B**) and cell image (**C**) as used for image analysis. The scale bars are 20 µm.

**Figure 3 pharmaceutics-13-00640-f003:**
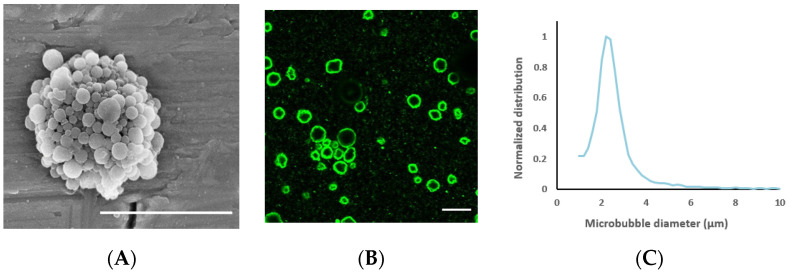
(**A**): Scanning electron microscopy image of a nanoparticle-loaded microbubble, demonstrating the morphology of the shell; the scale bar is 5 µm. Image reprinted with permission from [39]. (**B**) Confocal microscopy image of microbubbles to visualize nanoparticle loading by fluorescence (green); the scale bar is 10 µm. (**C**): Representative size distribution of the microbubbles.

**Figure 4 pharmaceutics-13-00640-f004:**
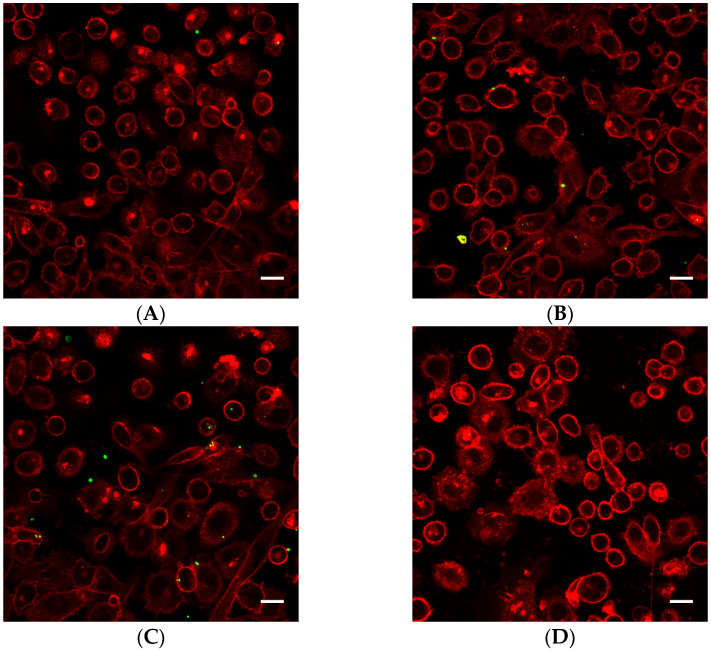
Representative overview confocal images of PC3 cells showing Cell Masks (red) and nanoparticles (green) both on the surface and inside the cells. In addition to the free nanoparticles, the cells were treated with (**A**) nanoparticle-loaded microbubbles but no ultrasound at 37 °C (control), (**B**) nanoparticle-loaded microbubbles and ultrasound at 4 °C, (**C**) nanoparticle-loaded microbubbles and ultrasound at 37 °C or (**D**) Sonazoid and ultrasound at 37 °C. The scale bars are 20 µm.

**Figure 5 pharmaceutics-13-00640-f005:**
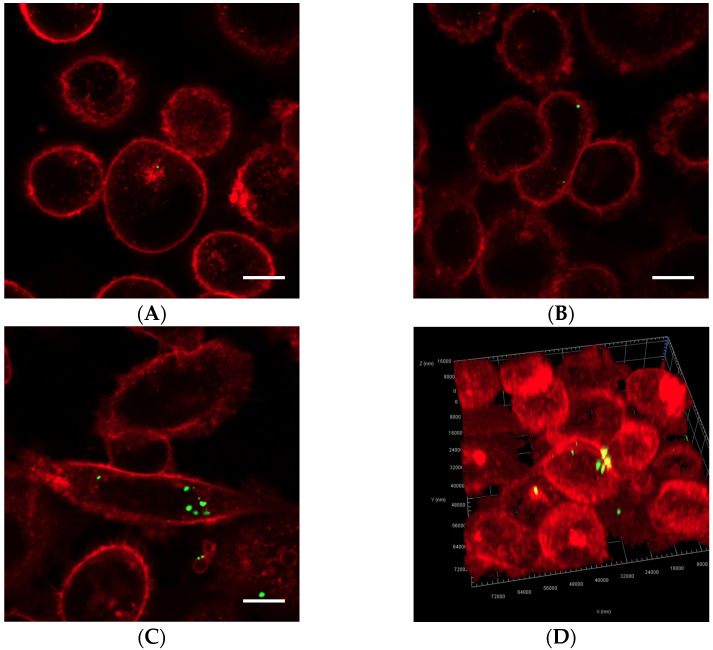
Representative confocal images of cells displaying Cell Masks (red) and nanoparticles (green). In addition to the free nanoparticles, the cells were treated with (**A**) nanoparticle-loaded microbubbles but no ultrasound at 37 °C (control), (**B**) Sonazoid and ultrasound at 37 °C or (**C**) nanoparticle-loaded microbubbles and ultrasound at 37 °C. Both individual nanoparticles and groups of a few can be observed, as well as larger nanoparticle clusters in C. The scale bars are 10 µm. (**D**) 3D image of a cell where the nanoparticles inside the cell (green) and on the cell surface (yellow) are visualized.

**Figure 6 pharmaceutics-13-00640-f006:**
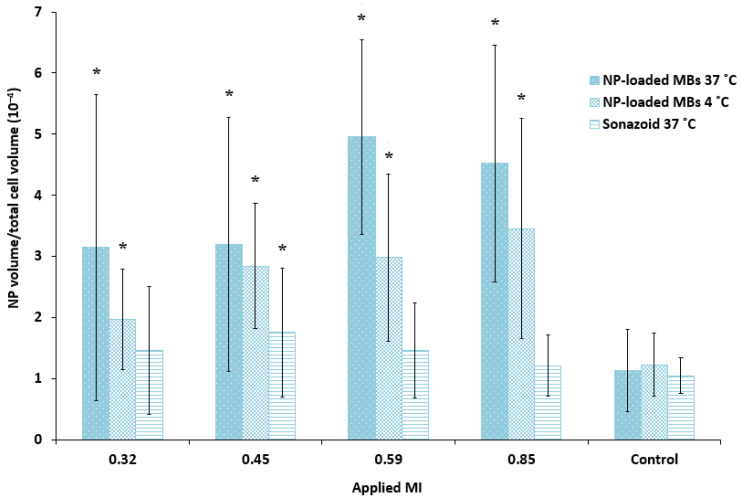
Quantitative comparison of the cellular uptake of nanoparticles (NP) after ultrasound treatment of various MIs using nanoparticle-loaded bubbles (NP-loaded MBs) at 37 °C and 4 °C, compared to the lipid Sonazoid microbubbles at 37 °C. Data show the total number of nanoparticle voxels relative to the total number of cell voxels in the z-stack (nanoparticles both inside cells and colocalized with the cell membrane are included). Averages and the standard deviation are shown from *n* = 17–18 z-stacks at 37 °C and *n* = 8–9 at 4 °C, and * indicates significant differences (*p* < 0.05) compared to the untreated control cells.

**Figure 7 pharmaceutics-13-00640-f007:**
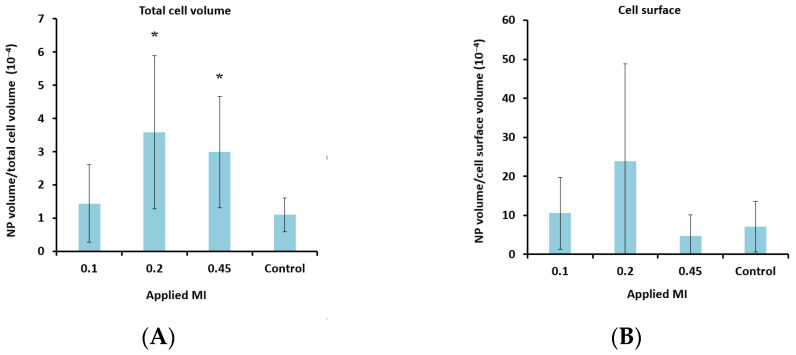
Effect of MI (**A** + **B**) and DC (**C** + **D**) on cellular uptake (**A** + **C**) and amount of nanoparticles (NP) colocalized with the cell membrane (**B** + **D**) after treatment with nanoparticle-loaded microbubbles and ultrasound. **A** + **C**: Data show the total number of nanoparticle voxels relative to the total number of cell voxels (nanoparticles both inside cells and colocalized with the cell membrane are included). **B** + **D**: Data show the total number of nanoparticle voxels relative to the cell surface voxels in the z-stack. Averages and the standard deviation are shown from *n* = 17–35 z-stacks for the treated cells and *n* = 11–13 for the untreated controls, and * indicates significant differences (*p* < 0.05) compared to the untreated control cells.

## Data Availability

Data is available upon request.

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
