# Peer review of "Sonoporation Using Nanoparticle-Loaded Microbubbles Increases Cellular Uptake of Nanoparticles Compared to Co-Incubation of Nanoparticles and Microbubbles"

_pharmaceutics, 2021, doi:10.3390/pharmaceutics13050640_

Round 1

Reviewer 1 Report

Snipstad et al. evaluated the cellular uptake difference between microbubbles and nanoparticle-stabilized microbubbles under sonication treatment. The study makes sense as the stabilized microbubbles have been previously reported (for example: DOI: 10.1002/anie.201509601). The current work is missing a few experiment to validate the design. 

  1. No characterization was provided for the polymer synthesis, microbubble fabrication, NP-stabilized microbubbles. This is a critical point that has to be addressed for reproducibility and quality control.
  2. How was these particles/bubbles labeled with a green fluorophore? Please elaborate the details and the purification/characterization on the labeling. 
  3. The uptake of these materials are a bit far from sufficient cellular uptake. Take Figure 4c as an example, only a few puncta were observed in one cell out of 4~5 cells in the field. This is indeed an improved uptake compared to Figure 4A,B, however, quite a low uptake in general. 
  4. All of these microscopic analysis only provided the evidence from one aspect. The result should be further validated by flow cytometry to provide the experimental evidence from a high-throughput perspective. The cell numbers being analyzed in these microscopy images were simply not enough.
  5. What is the endocytic mechanism of these particles/bubbles? Please provide experimental data on this. Suggested references to be followed for such experiments and included: DOI: 10.1021/acsnano.7b02044; DOI: 10.1021/acs.biomac.9b01073. 

Author Response

We thank the reviewers for their constructive feedback on how to improve the manuscript. Our responses and the corresponding changes that have been made to the manuscript are highlighted below in red. All changes in the manuscript, including some minor language edits, are done with track changes.

Reviewer 1:

Snipstad et al. evaluated the cellular uptake difference between microbubbles and nanoparticle-stabilized microbubbles under sonication treatment. The study makes sense as the stabilized microbubbles have been previously reported (for example: DOI: 10.1002/anie.201509601). The current work is missing a few experiment to validate the design. 

  1. No characterization was provided for the polymer synthesis, microbubble fabrication, NP-stabilized microbubbles. This is a critical point that has to be addressed for reproducibility and quality control.

The monomers for making nanoparticles were purchased as described in the materials and methods, page 3 line 113. A more detailed characterization of the polymeric nanoparticles has been done in several previous studies, a sentence about this has been included in page 3 line 124-126 with corresponding references. The microbubbles have also previously been characterized, a sentence about this was included on page 3 line 137-141, and page 11 line 386-388. An electron microscopy image and a confocal image to visualize morphology and nanoparticle loading of the microbubbles, along with their size distribution were included in a new figure 3, page 6 line 262-267. The methods section was updated accordingly, page 3 line 142-146.

  1. How was these particles/bubbles labeled with a green fluorophore? Please elaborate the details and the purification/characterization on the labeling. 

Labeling of the nanoparticles is done during nanoparticle synthesis. To make nanoparticles, an oil phase of the monomer containing the fluorescent dye is mixed with a water phase containing PEG, and sonicated to create a mini-emulsion where the oil droplets contain the hydrophobic fluorescent dye (NR668). The droplets are then allowed to polymerize, forming solid nanoparticles with the dye embedded in the polymer matrix. The dispersion is dialyzed to remove unreacted PEG and leftover dye. This was previously described on page 3 line 111-115, but we also included now that leftover dye is removed by dialysis on page 3 line 122. The dye NR668 was chosen because it was previously found to label the nanoparticles in a stable manner and to not leak out: https://pubmed.ncbi.nlm.nih.gov/27077940/ This reference was included in the methods before, but we have now included one more sentence to mention the motivation for why this specific dye was used, see page 3 line 116-117.

  1. The uptake of these materials are a bit far from sufficient cellular uptake. Take Figure 4c as an example, only a few puncta were observed in one cell out of 4~5 cells in the field. This is indeed an improved uptake compared to Figure 4A,B, however, quite a low uptake in general. 

We completely agree with the reviewer that the uptake is low, and that there is a large variation. The low uptake is one of the main findings and shows that sonoporation and cellular uptake of nanoparticles are challenging. This was already mentioned in the results section, but we have included a few sentences to make it clearer and to emphasize that the majority of cells did not contain nanoparticles on page 6 line 257-260 and page 8 line 310-311. The variation in cellular uptake was previously discussed on page 11 line 396-403; Since the MBs have a highly heterogeneous shell in terms of thickness and nanoparticles coverage, in combination with polydispersity within a batch, it is expected that different microbubbles within a batch will show variable behavior and thus variable cell interaction at the same ultrasound settings. This could explain the large variation in uptake observed between the individual cells. Making the microbubbles monodisperse to have a more homogeneous response to ultrasound treatment could probably lead to more effective drug delivery. Another option would be to optimize the number of microbubbles relative to the number of cells, which other groups have done. However, this highly variable and quite low number of cells being sonoporated was also reported by several other groups, since only the cells that are in contact with the bubble will be sonoporated. We included a couple of more sentences in the discussion on page 11 line 403-408. When using small molecules rather than these fairly large nanoparticles, the general trend in the literature seems to be that each cell receives more when the pore opens and the molecules diffuse in. This demonstrates the importance of optimizing both the type of microbubble and the type of ultrasound treatment to make delivery as efficient as possible, and we therefore expect that the current paper will be useful for the field.

  1. All of these microscopic analysis only provided the evidence from one aspect. The result should be further validated by flow cytometry to provide the experimental evidence from a high-throughput perspective. The cell numbers being analyzed in these microscopy images were simply not enough.

We agree that validation by flow cytometry would be a nice complement to the confocal imaging. However, with the setup that was used, that was unfortunately not feasible. The cells were cultured in CLINIcells, and because of the small focus of the ultrasound transducer, only a small area of the CLINIcell was exposed. When detaching all the cells in the CLINIcell for flow cytometry, the treated cells would only make up a tiny fraction of the total number of cells, making it difficult to distinguish between treated and untreated cells, and treated cells taking up nanoparticles. We did make an attempt of cutting out a small piece of the CLINIcell to only analyze the cells in the treated area, but this method was not robust enough and resulted in detachment of the cells. Another option would be to scan the transducer around to treat a large area, but our setup did not allow for that. Instead, we tried to compensate by taking 8-18 images per treated area. With each image containing 50-80 cells, the total number of cells analyzed is still fairly high, so we do not agree that the cell number analyzed is not enough. The number of cells per image is mentioned in the materials and methods page 5 line 207, and the number of images per treatment is given in the legends of figure 5 and 6 (now renamed to figure 6 and 7).

  1. What is the endocytic mechanism of these particles/bubbles? Please provide experimental data on this. Suggested references to be followed for such experiments and included: DOI: 10.1021/acsnano.7b02044; DOI: 10.1021/acs.biomac.9b01073. 

Endocytosis and degradation of the nanoparticles, including endocytosis mechanisms, have been described previously in a separate paper (no microbubbles involved) with prostate PC3 cells and rat brain endothelial RBE4 cells: https://pubmed.ncbi.nlm.nih.gov/26743777/

Endocytosis has also been studied for similar polymeric nanoparticles by other groups. However, in the current study, as can be seen from figure 5 (now figure 6), the delivery to cells is still high at 4 degrees Celsius when endocytosis is inhibited. This shows that the dominant mechanism for delivery to cells is not endocytosis, but rather sonoprinting/sonoporation as mentioned on page 12 line 434-435.

Reviewer 2 Report

The manuscript by Sofie Snipstad et al. describes the evaluation of the effect of ultrasound application on nanoparticle uptake by cells in the presence of microbubbles. The main focus of the work is to compare the effect on nanoparticle internalization by the cells when the nanoparticles are either attached to the microbubble surface or simply mixed with them without any chemical or physical interaction. The experiments were designed well (especially important for the experimental setup from the acoustics point of view) and the results are shown in a clear way. The work is relevant, although the idea behind it is not particularly novel (which is my main concern regarding the publication of this manuscript). Could the authors emphasize the novelty of the work and /or methods used compared to other available works regarding nanoparticle sonoprinting/sonoporation?

As additional comments:

  1. On page 4, line 183, the authors mention Figure 3C, when I believe it should be Figure 2C.
  2. For all the results in Figures 5 and 6, multiple statistical comparisons are shown. However, the authors state in the Experimental section that they used “two-sample student t-test”, without any mention to any correction for multiple comparisons.

Author Response

We thank the reviewers for their constructive feedback on how to improve the manuscript. Our responses and the corresponding changes that have been made to the manuscript are highlighted below in red. All changes in the manuscript, including some minor language edits, are done with track changes.

Reviewer 2:

The manuscript by Sofie Snipstad et al. describes the evaluation of the effect of ultrasound application on nanoparticle uptake by cells in the presence of microbubbles. The main focus of the work is to compare the effect on nanoparticle internalization by the cells when the nanoparticles are either attached to the microbubble surface or simply mixed with them without any chemical or physical interaction. The experiments were designed well (especially important for the experimental setup from the acoustics point of view) and the results are shown in a clear way. The work is relevant, although the idea behind it is not particularly novel (which is my main concern regarding the publication of this manuscript). Could the authors emphasize the novelty of the work and /or methods used compared to other available works regarding nanoparticle sonoprinting/sonoporation?

The main findings are that using this unique nanoparticle-loaded microbubble we confirm that successful uptake of nanoparticles in cells in vitro requires proximity between nanoparticles, oscillating microbubbles and the cell surface. This is the first time such in vitro experiments have been published for these specific microbubbles. Confirming the results from de Cock and colleagues who used another nanoparticle-microbubble complex (where liposomes did not form the shell of the microbbles), are important for the general understanding of how sonoporation contributes to cellular uptake of nanoparticles. The methods we have used are not novel compared to previous experiments regarding nanoparticles sonoprinting/sonoporation, as flow cytometry and confocal microscopy are the commonly used methods. We have emphasized the novelty in the introduction, page 2 line 65-83.

As additional comments:

  1. On page 4, line 183, the authors mention Figure 3C, when I believe it should be Figure 2C.

We thank the reviewer for pointing this out, the typo has been corrected, page 5 line 216.

  1. For all the results in Figures 5 and 6, multiple statistical comparisons are shown. However, the authors state in the Experimental section that they used “two-sample student t-test”, without any mention to any correction for multiple comparisons.

We thank the reviewer for pointing this out. The analysis has now been redone, including correction for multiple comparisons. Details of the updated analysis are described on page 5 line 224-230. The bar plots in figure 5 and 6 (now 6 and 7) were updated accordingly. To make sure that everything would be correct we consulted a statistician who is now acknowledged for advice in page 13 line 496. Briefly, the data were first tested for normality with multiple Shapiro – Wilk tests. For some groups, the data were not univariate normal, hence a non-parametric method rather than a t-test was used. Pairwise Wilcoxon rank sum tests (also called Mann-Whitney U tests) were performed to test for significance. A significance level of 0.05 was used in all hypothesis tests. To correct for multiple comparisons, the False Discovery Rate was used to adjust the p—values.

Reviewer 3 Report

I have thoroughly read the manuscript titled, “Sonoporation Using Nanoparticle-Loaded Microbubbles Increases Cellular Uptake of Nanoparticles Compared to Co-Incubation of Nanoparticles and Microbubbles”. I have also read previous publications by the authors where the same studies and nanoparticles have been synthesized with same types of experiments. For example below are only two recent publications and it is not clear where the current manuscript presents significant and novel findings compared to these types of publications. Furthermore, these publications report in vivo results using the same system and types of experiments so it is highly unlikely that the current in vitro experiments will be helpful in providing insights when in vivo experiments have already been published.

Previous Publications:

Sonopermeation Enhances Uptake and Therapeutic Effect of Free and Encapsulated Cabazitaxel

Cabazitaxel-loaded Poly(2-ethylbutyl cyanoacrylate) nanoparticles improve treatment efficacy in a patient derived breast cancer xenograft

There are also major important results missing (see below).

  • Detailed characterization of the microbubbles stabilized by nanoparticles should be provided such as results for size distribution, zeta potential and electron microscopy for morphology instead of just reporting values for the size and charge. I have seen these important results already published previously.
  • Detailed experiments and results related to bubbles and nanoparticles (not only bubbles stabilized by nanoparticles) (as shown in Figure 1) with cells should be provided along with the current results to show that the nanoparticle-stabilized microbubbles can be used to enhance uptake of particles by cells. Such types of results are also previously published from in vivo
  • Figures 3 and 4 are not convincing to show that cell internalization is enhanced with nanoparticle-stabilized bubbles and nanoparticles are very sparse.
  • There is too much variability in the results in Figures 5 and 6 to show cellular uptake and the standard deviation is very large showing large variability. The results are significant only because the results from controls and bubbles only have lower standard deviation and lower mean values.

Instead of only mentioning their own work, the authors should mention other types of similar nanoparticles and nanoparticle clusters or nanoparticles coating other nanoparticles that have been synthesized for enhancing therapy. The authors should introduce the use of such nanoparticles to show they have previously been used for effective treatment and therapy (see for example below publication).

  • Fernandes, D. A., Fernandes, D. D., Li, Y., Wang, Y., Zhang, Z., Rousseau, D. r., . . . Kolios, M. C. (2016). Synthesis of stable multifunctional perfluorocarbon nanoemulsions for cancer therapy and imaging. Langmuir, 32(42), 10870-10880.

The conclusions section is very short. Some important points missing are to explain how the results presented are important in the context of the overall picture for improving therapy and what are important next steps. The same statements should not be repeated from the previous sections and abstract.

Author Response

We thank the reviewers for their constructive feedback on how to improve the manuscript. Our responses and the corresponding changes that have been made to the manuscript are highlighted below in red. All changes in the manuscript, including some minor language edits, are done with track changes.

Reviewer 3:

I have thoroughly read the manuscript titled, “Sonoporation Using Nanoparticle-Loaded Microbubbles Increases Cellular Uptake of Nanoparticles Compared to Co-Incubation of Nanoparticles and Microbubbles”. I have also read previous publications by the authors where the same studies and nanoparticles have been synthesized with same types of experiments. For example below are only two recent publications and it is not clear where the current manuscript presents significant and novel findings compared to these types of publications. Furthermore, these publications report in vivo results using the same system and types of experiments so it is highly unlikely that the current in vitro experiments will be helpful in providing insights when in vivo experiments have already been published.

As commented by the reviewer, these nanoparticle-loaded microbubbles have previously only been used for in vivo experiments. The bubbles have been used for enhancing delivery of nanoparticles to both tumors in mice and across the blood brain barrier. Previous papers demonstrate uptake in tumor tissue, microdistribution of nanoparticles in tissue, and therapeutic effect in vivo. After reading the work from de Cock et al about nanoparticle-loaded lipid microbubbles being effective also for in vitro drug delivery to cells, we wanted to investigate if that was the case also for our nanoparticles loaded protein microbubbles. This is the first time such in vitro experiments have been published for these specific microbubbles. Our results confirm the findings by the de Cock and colleagues, and show that when aiming for delivery of larger substances such as nanoparticles, the interaction and proximity between nanoparticles, microbubbles and cells is highly important. We have emphasized the novelty in the introduction and how this in vitro study is connected to previously published in vivo studies, page 2 line 65-83. We also varied the ultrasound parameters to understand how that affected delivery to cells. Such detailed insight into cellular delivery has not been possible with the previously performed in vivo experiments. This gives us complementary information to the in vivo experiments, and allows us to understand more about how the system works.

Previous Publications:

Sonopermeation Enhances Uptake and Therapeutic Effect of Free and Encapsulated Cabazitaxel

Cabazitaxel-loaded Poly(2-ethylbutyl cyanoacrylate) nanoparticles improve treatment efficacy in a patient derived breast cancer xenograft

There are also major important results missing (see below).

  • Detailed characterization of the microbubbles stabilized by nanoparticles should be provided such as results for size distribution, zeta potential and electron microscopy for morphology instead of just reporting values for the size and charge. I have seen these important results already published previously.

Currently in the results section, we report the size, polydispersity index and zeta potential of the nanoparticles, and the average size and concentration of the microbubbles. In previous publications we included electron microscopy images to show morphology of the nanoparticles and microbubbles, and also detailed description of size distribution, stability in vitro and in vivo, reproducibility, and acoustic properties. A new a sentence referring to previous characterization is included in the materials and methods part, page 3 line 137-141. We have included a new figure showing both a confocal image and an electron microscopy image of the microbubbles to show their morphology and nanoparticle loading on the microbubble surface, in addition to their size distribution, which is the new figure 3. Corresponding information has been included in the materials and methods page 3 line 141-146, and in the results section page 6 line 246-248. Figures 3, 4, 5 and 6 have therefore been renamed to 4, 5, 6 and 7 both in figure legends and throughout the text.

  • Detailed experiments and results related to bubbles and nanoparticles (not only bubbles stabilized by nanoparticles) (as shown in Figure 1) with cells should be provided along with the current results to show that the nanoparticle-stabilized microbubbles can be used to enhance uptake of particles by cells. Such types of results are also previously published from in vivo

We apologize if we misunderstood the question. The comparison between the nanoparticle-loaded microbubble and the case with bubbles and nanoparticles separately was already shown figure 3 and 4, and quantified in figure 5 (now renamed to figure 4, 5 and 6). As shown, the delivery to cells is more efficient when the nanoparticles are loaded onto the bubbles. The lipid microbubbles Sonazoid were used for comparison. If the reviewer means comparison to exactly the same type of bubble both with and without the nanoparticles, that is something we have also tried previously. However, the microbubbles will not be stable if only the protein is used without adding any nanoparticles, the combination of protein and nanoparticles is needed to form stable bubbles, as mentioned in page 3 line 132-133. Hence, that kind of direct comparison is unfortunately not possible with this specific type of nanoparticle-loaded microbubble.

  • Figures 3 and 4 are not convincing to show that cell internalization is enhanced with nanoparticle-stabilized bubbles and nanoparticles are very sparse.

We agree with the reviewer, and the images in figure 3 and 4 (now 4 and 5) were used as representative examples to give a visual impression on the fact that nanoparticles had accumulated in very few cells. This demonstrates the challenge in successful cellular uptake of nanoparticles caused by sonoporation. The delivery/uptake was quantified in 8-35 images each containing 50-80 cells, as shown in figure 5 and 6 (now 6 and 7). Please see also our reply to question 4 from reviewer 1 on the same topic.

  • There is too much variability in the results in Figures 5 and 6 to show cellular uptake and the standard deviation is very large showing large variability. The results are significant only because the results from controls and bubbles only have lower standard deviation and lower mean values.

We agree with the reviewer that the uptake is low, and that there is a large variation. This was already mentioned in the results section, but we have included one more sentence to make it more clear on page 6 line 257-260 and page 8 line 310-311. Possible reasons were previously discussed on page 11 line 396-402, and updated with line 403-408, as explained in the response to question 3 from reviewer 1. However, this highly variable and quite low number of cells being sonoporated was also reported by other groups. This demonstrates the importance of optimizing both the type of microbubble and the type of ultrasound treatment to make delivery as efficient as possible, and we therefore expect that the current paper will be useful for the scientific community working in this field.

Instead of only mentioning their own work, the authors should mention other types of similar nanoparticles and nanoparticle clusters or nanoparticles coating other nanoparticles that have been synthesized for enhancing therapy. The authors should introduce the use of such nanoparticles to show they have previously been used for effective treatment and therapy (see for example below publication).

  • Fernandes, D. A., Fernandes, D. D., Li, Y., Wang, Y., Zhang, Z., Rousseau, D. r., . . . Kolios, M. C. (2016). Synthesis of stable multifunctional perfluorocarbon nanoemulsions for cancer therapy and imaging. Langmuir, 32(42), 10870-10880.

The introduction and discussion contain references to what we consider as the most relevant studies demonstrating various nanoparticle-loaded microbubbles for drug delivery (Burke et al, de Cock et al, Roovers et al, Vandenbroucke et al). The suggested reference to nanoemulsions and a couple of more have been included in the discussion on page 12 line 457, where we previously mentioned that these types of novel nanodroplets/nanobubbles/nanoemulsions could be promising.

The conclusions section is very short. Some important points missing are to explain how the results presented are important in the context of the overall picture for improving therapy and what are important next steps. The same statements should not be repeated from the previous sections and abstract.

A couple of sentences have been added to the conclusion to emphasize possible applications in the overall picture of how the reported findings could be used to improve therapy, see page 13 line 474-483.

Round 2

Reviewer 1 Report

The authors have tried to address the questions from previous reviewers. 

Reviewer 2 Report

The authors have succesfully addressed my concerns.